# Integrating co-design into formative research for a SBCC entry-point platform for nutrition-sensitive social protection programs in low -and middle-income country settings

Tahir Turk[1,2]*, Rina Paul[3], Nilofer Fatimi Safdar[1], Nadia Shah[1], Syed Mahbubul Alam[4], Sohel Reza Choudhury[3], Kashif Shafique[1]*, Zaeema Ahmer[1], Anthony Wenndt[5,6,7], Hamidullah Khan Babar[5,6,7], Tannaza Sadaf[5,6,7], Moniruzzaman Bipul[5,6,7]

**1** School of Public Health, Dow University of Health Sciences, Karachi, Pakistan, **2** Communication Partners International (CPI), Wamberal, New South Wales, Australia, **3** Department of Epidemiology and Research, National Heart Foundation Hospital and Research Institute, Dhaka, Bangladesh, **4** Center for Law and Policy Affairs, Dhaka, Bangladesh, **5** Global Alliance for Improved Nutrition (GAIN), Washington DC, USA, **6** Global Alliance for Improved Nutrition (GAIN), Islamabad, Pakistan, **7** Global Alliance for Improved Nutrition (GAIN), Dhaka, Bangladesh

* tturk@cpimail.net

## Abstract

### Background

Achieving adequate nutrition for vulnerable populations is an objective of the Sustainable Development Goals. Nutrition-sensitive social protection programs, including those that promote nutrition through Social and Behaviour Change Communication (SBCC), have the potential to reduce malnutrition and provide social supports to those most in need. Country-level needs assessments can clarify key issues. When supported by co-design approaches, program formative research may provide more culturally contextualised SBCC for the improved delivery of nutrition social protection to vulnerable groups. This formative study from Pakistan and Bangladesh, integrated co-design to more fully explore program beneficiary knowledge, attitudes and perceptions toward nutrition social protection to inform the design of SBCC key messages and an entry-point platform to ensure effective message dissemination.

### Methods

Qualitative formative research was conducted to support findings from a systematic review. Thirty semi-structured interviews with program stakeholders and 12 focus group discussions (134 participants) were conducted with program beneficiaries in Bangladesh and Pakistan. Co-design sessions supplemented the needs assessment protocol. A COREQ checklist ensured best practice approaches in research design, analysis and reporting. NVIVO 2023 qualitative software supported the thematic analysis.

**Data availability statement:** The data and documentation for this article are published via the Qualitative Data Repository: https://doi.org/10.5064/F6M0T6GO. All documentation files are freely accessible under the Creative Commons Attribution-Share Alike 4.0 license, while data files are accessible without restrictions for all registered QDR users under our Standard Download Agreement (https://qdr.syr.edu/discover/standarddownload) in line with best human participant research practices. The data files are accessible under QDR's Standard Access conditions described here: https://qdr.syr.edu/policies/qdr-standard-access-conditions.

**Funding:** This study was produced through the Global Alliance for Improved Nutrition (GAIN) Nourishing Food Pathways programme which is jointly funded by the German Federal Ministry for Economic Cooperation and Development; the Ministry of Foreign Affairs of the Netherlands; the European Union; the government of Canada through Global Affairs Canada; Irish Aid through the Development Cooperation and Africa Division (DCAD); and the Swiss Agency for Development and Cooperation (SDC) of the Federal Department of Foreign Affairs (FDFA). The funders had no role in study design, data collection and analysis, decision to publish, or preparation of the manuscript.

**Competing interests:** Turk T, Shafik K, Choudhury S, Safdar N, and Mahbub-Alum S. are independent consultants to GAIN. Wenndt AJ, Babar HK, Sadaf T, and Bipul M work for GAIN as full-time employees. Other authors declare that they have no competing interest.

## Results

Four organising themes were identified: 1. Barriers to Program Engagement, 2. Opportunities for Program Improvement, 3. Knowledge Attitudes and Practices, and 4. Target Groups, Messaging and SBCC Entry-Points, with 21 sub-themes emerging under the four organising themes. Main barriers related to resource constraints and maladministration of SPPs while opportunities identified greater integration of cash transfers with nutritious food provision, increased engagement with key influencers in vulnerable communities, and identification of culturally nuanced messages with dissemination through preferred channels. Integrating co-design sessions provided greater ownership, participation and engagement by program beneficiaries and more pragmatic SBCC solutions to challenges identified.

## Conclusion

The needs assessments and integrated co-design sessions highlighted the benefits of close consultation with program beneficiaries in the design of culturally appropriate SBCC interventions to support nutrition-sensitive social protection programs. A SBCC entry-point platform was developed from participant recommendations to provide options for programmers on message designs, advocacy approaches and dissemination channels with the approaches applicable for a number of low -and middle-income countries where malnutrition is a major challenge.

## Introduction

Malnutrition remains a concern for vulnerable populations globally, and improving nutrition is critical in achieving the Sustainable Development Goals (SDGs). Globally, millions of people depend on social protection programs (SPPs) to meet their basic needs while safeguarding their food security. While "nutrition-specific" interventions can address the immediate determinants of child and maternal nutrition and development—adequate food and nutrient intake, feeding, caregiving and parenting practices, and lower the burden of infectious diseases, "nutrition-sensitive" interventions for social protection are those that address the underlying determinants of malnutrition, and often aim to promote behaviours conducive to nutrition through social and behaviour change communication (SBCC) while incorporating specific nutrition goals and actions. A growing body of evidence shows that nutrition-sensitive programs that include SBCC can lead to improved maternal and infant health and nutrition outcomes [1,2]. SBCC aims to improve behaviours of individuals through dialogues and processes that strengthen social contexts and systems [3].

Nutrition-sensitive SBCC interventions can also improve feeding practices for infants and young children early initiation while emphasizing the importance of exclusive breastfeeding and adequate complementary feeding practices, along with water and sanitation hygiene practices [4]. Cash transfer programs in social protection focus strongly on outcomes linked with the SDGs and are therefore important as

they provide a crucial safety-net to children and their caregivers to alleviate poverty and address the key determinants of malnutrition [5]. However, despite the potential for integrating nutrition-sensitive SBCC interventions into SPPs, there is mixed evidence on the extent to which these interventions may deliver improved nutrition outcomes. To date, systematic reviews and meta-analyses have tended to focus on the importance of cash transfer programs for improvement in nutritional outcomes [6,7] or the added impact of SBCC interventions in communities to achieve optimal indicators for nutrition [6,8]. Yet to be answered systematically is how nutrition-sensitive SBCC can best be integrated within social protection programs to achieve the optimal behavioural impact in the context of low -and middle-income countries like Bangladesh and Pakistan [9]. Emerging evidence identifies the importance of integrating co-design approaches with those targeted by SBCC interventions to ensure the optimal development of SBCC program components [10,11].

The objective of this needs assessment study was to pilot an improved formative research method to integrate co-design approaches with program beneficiaries to enhance the findings of a systematic review [12]. Study objectives included the following: 1. Identify stakeholder and beneficiary current knowledge, attitudes and practices (KAP) in relation to nutrition-sensitive SPPs in Bangladesh and Pakistan; 2. Identify opportunities for the SBCC program moving forward – key messages, actors and communication channels and SBCC entry points for continuous program improvement; 3. Identify optimal monitoring and evaluation components of the SBCC nutrition-sensitive SPPs, including inputs, outputs, outcomes and impact.

## Methods

A landscape analysis and systematic review of the literature was conducted exploring the effectiveness of nutrition-sensitive SBCC approaches in the context of SPPs in Bangladesh and Pakistan, with the review methodology and findings reported [13] elsewhere. To provide a more culturally nuanced understanding of the nutrition social protection issues, to supplement the systematic review, a needs assessment was conducted with program stakeholders and beneficiaries in both countries. The recruitment of participants for this study commenced on 2nd August (Pakistan) and 20th November (Bangladesh) with the participant recruitment period ending in both countries on 14th December 2023. Co-design workshops of approximately 30 mins each were incorporated within research approaches. An interpretive descriptive method was adopted to the qualitative research to explore attitudes and opinions of supply-side and demand-side participants most invested in achieving successful program outcomes [14]. Semi-structured interviews (SSIs) were the preferred approach for working with SPP stakeholders given these key informants are often in senior professional roles with busy schedules. The approach is effective in gaining insights into problems that are not immediately perceptible and provides flexibility to probe for specific organisational and program challenges as well as opportunities for program improvement [15].

Qualitative approaches with program beneficiaries from the two countries took the form of focus group discussions (FGDs) as the approach provides for a non-threatening environment in which a moderator can explore knowledge, attitudes and practices (KAP) in relation to nutrition-sensitive SPPs with like-minded groups of 8–12 community participants. The methodological guidelines for FGDs included training of highly skilled, local moderators (MPHs, PhDs), the use of a standardised discussion agenda exploring a range of KAP indicators; probing with all respondents, avoiding any 'leading' of the groups, and close monitoring of the groups to identify potential positivity bias and dominant respondent bias [16]. Given the desire for greater integration of participatory approaches with the FGDs, a workshop session of Co-Design/Co-Creation [17] of approximately 30 mins duration, was incorporated into the group moderation to allow for greater collaboration from community members most affected by poor nutrition outcomes and multiple vulnerabilities. A 32-item COREQ (COnsolidated criteria for REporting Qualitative research) checklist [18] was used to ensure best practice approaches in the qualitative study design for conducting the SSIs and FGDs (see Supplementary Files).

Data analysis utilised Grounded Theory given the approaches ability to develop SBCC theory 'grounded' in data that has been systematically collected and analysed [19] and its ability to uncover issues related to social relationships/

social processes and group behaviours [20]. Furthermore, thematic analysis was adopted for the analysis as the process allows for the formation of themes, in this case related to nutrition KAP, from participant transcripts and responses, to open-ended questions on a range of issues of interest. NVIVO 14 data analysis software was used to support the thematic analysis with the systematic review of secondary data sources supporting in-field findings. Ethics approval for the research was provided by the National Heart Foundation Hospital and Research Institute (Bangladesh) IRB clearance NHFH&R1 4/19/7-AD/2369, and Dow University of Health Sciences (Pakistan), Internal Review Board: IRB-3123/DUHS/Approval/2023/276.

## Discussion agenda

Standardized discussion agenda were developed for the SSIs and FGDs to explore a number of issues related to nutrition-sensitive SBCC in the context of SPPs in Bangladesh and Pakistan. Items explored knowledge, attitudes and practices (KAP) of program beneficiaries in relation to SPPs and access to food supplementation. Informed consent was requested of participants to take part in the study with an item exploring stakeholder *Source Credibility* such as their qualifications and experience. Next, were items on *Perceived Challenges* to vulnerable groups accessing SPPs and improved nutrition outcomes, followed by *Perceived Opportunities* for improved communication for behavioural change within the SPPs to support improved nutrition. Probing was then conducted on anticipated *Target Groups* for the nutrition-sensitive SBCC interventions and perceptions on current *SPPs Capacity* to support beneficiaries to better attain nutritious diets. Follow-up items related to anticipated *Messaging/Creative Approaches* for SBCC and *Entry-Points* and *Dissemination Channels* for SBCC. Finally, items explored perceptions of *SPP Branding* and opportunities for incorporating *Behavioural Incentives*, with an open-ended question on any *Other Issues* deemed pertinent to the investigation, before closing.

The discussion agenda for the FGDs followed a similar line of enquiry to the SSIs but included a workshop co-design session. Key issues emanating from a barrier and benefit analysis in the initial stage of the FGDs, framed topics that were explored in greater depth through the co-design workshop sessions. A screener instrument was developed to screen FGD participants for relevant gender, age, SES, and location criteria, and to also allocate 50% quotas for those already enrolled in SPPs, and those not currently enrolled. The screener also required participants to sign a clearance for *Informed Consent* to take part in the study.

Specific items for the FGDs included a brief warm-up session where participants were asked about their family status and one thing they *Liked, Didn't Like and Want to Change* in their communities. A subsequent item that tapped into mentions of nutrition and welfare by beneficiaries identified the topic for discussion, with a question on who had heard of *SPPs in your area?* Following items explored participants' *Perceptions of SPPs,* for those who were enrolled as well as those currently not enrolled in the programs, and their reasons for seeking/or not seeking SPPs support. Another item explored participants knowledge of SPPs, including the names of the programs and the services delivered. Another item explored with those currently enrolled in an SPP, how they were made *Aware of the Program,* with probing on specific message sources. Next, was an item on the *Main Challenges or Barriers* as well as possible *Benefits* to vulnerable people accessing SPPs with the issues arising used as topic themes for the *Co-Design Workshop.* Exploration during the co-design session examined the key barriers and suggestions on how the barriers could be best addressed. This included probing on specific *Messages/Information* that could be provided, including probing on the preferred *Message Sources* and the predominant *Dissemination Channels* for participants to better access SPPs and nutrition benefits. Message *Entry-Points* were also explored to identify participants perceptions on the specific times SPP, and aligned nutrition messages should be delivered to support behavioural change. Two additional FGDs incorporating workshop sessions were conducted with SPP front-line field workers in Bangladesh, with the discussion agenda skewed to address issues and key topics related to front-line field worker perceptions of their beneficiaries SBCC needs and wants.

## Data collection

Male and female moderators were trained on how to use the discussion agenda to conduct the SSIs and FGDS. The discussion agenda was translated into local Bengali and Urdu languages and dialects by qualified professionals. Pilot FGDs were conducted in Bangladesh and Pakistan to trial the protocol with minor amendments made on the administration of the study. A total of 30 SSIs were conducted in Bangladesh and Pakistan: Twenty SSIs were conducted with supply-side key informants in Bangladesh, comprising senior Government officials involved with SPP administration – Health, Social Welfare, Women's Affairs, Food and Agriculture; International non-Governmental Organisations (INGOs) – World Food Program (WFP), UNICEF, Global Alliance for Improved Nutrition (GAIN), WaterAid, CARE and the International Food Policy Research Institute. National NGO stakeholders consulted included BRAC and the Bangladesh Center for Communication Programs (BCCP). Ten SSIs were also conducted in Pakistan with Government and donor funded SPPs including the Benazir Income Support Programme (BISP), Integrated Reproductive, Maternal, Newborn & Child Health (IRMNCH) Nutrition Program, the Social Welfare Department, and Aga Khan and Ziauddin Universities; INGOs – World Food Program (WFP), Global SUN Civil Society Network (Nutrition International), and Allah Walay Trust NGO.

A total of 12 FGDs were also conducted in Bangladesh and Pakistan, totalling 134 demand-side participants. Six FGDs were conducted in Bangladesh comprised of two groups of women, and two groups of men aged 19–34 years, and one group of men and one group of women aged 35–64 years respectively, with inclusion criteria for extreme poverty (SES-CD classifications extrapolated from the most recent census) and rural and urban geographic locations. Additionally, two groups of women and two groups of men of similar classifications were also recruited in urban areas of the country. An additional two FGDs of front line SPP field workers were also conducted – one in rural and one in an urban area of the country with FGDs in Bangladesh totalling 84 participants. Four FGDs were conducted in rural and urban areas of Pakistan with adult (18+years) females and males (two groups each) totalling 50 participants. A standardised screening tool and ethics clearance was used to recruit participants in both countries, with FGD participants required to sign consent forms prior to attending groups while key informant interviews required verbal consent prior to the interviews commencing.

## Data analysis

Audio recordings from the SSIs and FGDs were professionally transcribed into English from Bengali and Urdu languages with any identifying characteristics removed in subsequent iterations of the analysis. Transcripts of the SSIs and FGDs accounted for 244 pages of Word documents for coding and thematic analysis. The analysis was conducted in a number of iterative stages to build nutrition-sensitive SBCC theory from the data, that was both reflective and reflexive [21]. Anonymized transcripts were uploaded into NVivo Version 14 Plus (QSR International) to enable organized retrieval of the data for coding.

Thematic content analysis was used to inductively analyse the transcripts by the primary reviewer – TT independently coded the transcripts and established organising themes and sub-themes based on SBCC strategic planning needs, while RP from Bangladesh, and NFS from Pakistan independently reviewed the organising themes and sub-themes as second and third independent reviewers. Any revisions recommended by the reviewers were subsequently discussed among the group until agreement on the key themes and sub-themes was achieved. The findings from the SSIs and FGDs were later triangulated with findings from the aforementioned systematic review to identify any convergence, corroboration or discrepancies, while retaining internal/external validity and reliability [22].

## Results

Findings from the SSIs and FGDs identified a number of similar themes and sub-themes emanating from the thematic analyses. A thematic analysis coding table identified four organising themes, and 21 sub-themes emerging under the overarching theme under investigation – Nutrition Sensitive SBCC in Social Protection (see Fig 1).

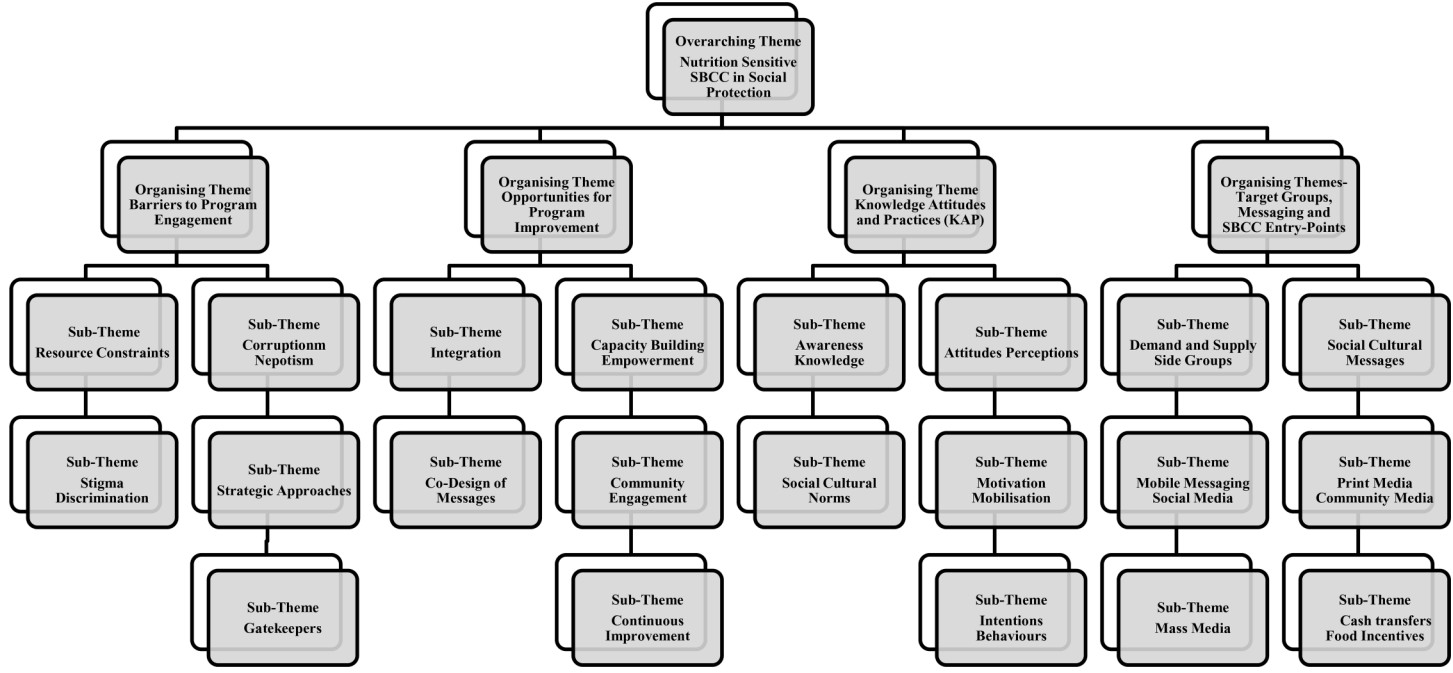

**Fig 1. Thematic Analysis Coding Table: Nutrition-Sensitive SBCC in Social Protection Programs in Bangladesh and Pakistan.**

The four organising themes related to *Barriers to Program Engagement, Opportunities for Program Improvement, Knowledge Attitudes and Practices (KAP),* and *Target Groups, Messaging and SBCC Entry-Points.* Of the 21 sub-themes, data analysis identified the most predominant theme identified by stakeholders and a number of beneficiaries within the *Barrier's* organising theme was *Resource Constraints,* with this issue achieving data saturation before the SSI and FGD data analysis was completed. The most commonly cited resource constraints included the insufficient level of financial support/food support provided by most SPPs and the inequitable allocation of placements within the programs, which greatly exceeded supply. The second most cited barrier, which also achieved data-saturation before completion of the analysis, was *Corruption and Nepotism* within the administration of SPPs. Attitudes toward maladministration of SPPs were prevalent in both countries. Many vulnerable community members who were not able to enrol into SPPs had poorer attitudes and perceptions toward the selection processes.

The third most cited barrier was the *Stigma and Discrimination* sub-theme, where beneficiaries claimed they had been humiliated through the process of applying for aid, or feared being labelled as destitute by other community members. A fourth sub-theme to emerge was the lack of strategic approaches, which prevented the attainment of well designed, evidence-based SBCC programs with the collaboration of all stakeholders Other barriers mentioned by beneficiaries and stakeholders included poverty, widespread inequity, and gender issues including 'gatekeepers', predominantly the husbands of women who impeded their wives ability to successfully engage in social protection cash transfer and food programs as a result of distrust of any programs which may take their wives out of the household. Other sub-themes emerging that were seen to hinder program uptake were policy inertia, low literacy, substance abuse, and the distances and time required to travel to SPP recruitment offices/distribution hubs and access points (see Table 1. Barriers to Program Engagement).

When examining the *Opportunities* organising theme for program improvement, several sub-themes emerged. The first is the sub-theme of *Integration* of nutrition-sensitive SBCC into SPPs which was identified as one of the study objectives.

**Table 1. Organising Theme – Barriers to Program Engagement.**

| Sub-Theme | Feedback |
|---|---|
| Resource Constraints | "Another thing that can be done is the amount of support they give; that is insufficient. But still then, whatever they are getting, access is quite a challenge. If you can increase the support in terms of money, in terms of numbers, in terms of amount, or volume, then people will benefit. Because the crisis in terms of prices is quite high." *Senior Director, HNPP, Health and Humanitarian Crisis Management BRAC Bangladesh.* "The hardest issue is inflation and food is expensive. What should we do; we make our children skip meals, but we have to pay bills." *Adult-Rural-Female-C-D SES, Sheikhupura Village, Pakistan.* "They need some kind of financial assistance or incentives to be part of the behavioral change campaigns. Otherwise, if you ask them to eat fruits or meat or lentils, they will not have money. They will ask to eat it, but money is needed." *Director General, Government Cash Transfer Program, Pakistan.* |
| Corruption/ Nepotism | "Those who have some connections in the office, they are given priorities. Only those get in the queue for enrolment." *Adult-Rural-Female-C-D SES, Sheikhupura Village, Pakistan.* "When you have limited resources, who will you choose? Suppose you have resources for three people then you choose the bottom three. But the problem occurs in choosing. When people have political connections, the upper three will get selected." *Director, Health and Humanitarian Crisis Program NGO, Bangladesh.* |
| Stigma/ Discrimination | "They gave loans to poor people up to PR500,000 for starting their own business, but I was humiliated so many times that I decided not to pursue them." *Adult-Urban-Male-C-D SES, Johar Town, Lahore, Pakistan.* |
| Strategic Approaches | "There is no comprehensive strategy for SBC. It is recommended that the government and its partners collaborate to develop a comprehensive SBC strategy. This strategy should be informed by evidence-based approaches and designed to effectively communicate key nutrition messages to beneficiaries." *Senior Advocacy Advisor, INGO SPP, Pakistan.* "I'm not sure about the policy behind it, but these programs are not advertised well. There is no promotion, and messages thus do not reach the people" *Program Officer, Nutrition, Integrated Reproductive, Maternal, Newborn & Child Health, Government of Pakistan.* |
| Gatekeepers | "It is hard for women to visit. Females want to get enrolled, but their husbands and family discourage them due to distrust of the program. They tell their women 'it is a useless program; you will not get the money'. It's a futile exercise." *Adult-Rural-Male-C-D SES, Sheikhupura Village, Pakistan.* |

Feedback from the supply and demand sides identified that a number of food programs are already being trialled in both countries, with opportunities to integrate approaches more fully with cash transfers.

Second, are opportunities for *Capacity Building/Empowerment* of program beneficiaries and other community members, including key influencers, with the approaches potential to build ownership and social mobilization of the whole community. Next, is the potential for *Co-Design/Co-Creation* which was articulated through the participatory workshop exercises embarked upon with program beneficiaries as a component of the FGDs. The approach assumed that it's those who are most vulnerable who may also be best placed to express their specific needs and wants in relation to SBCC, nutrition and social protection. Next, are opportunities for *Improved Community Engagement* to ensure that those in most need are provided for in safety-nets, and any gaps or leakages can be addressed in conjunction with empowered community representatives and groups.

Last, are *Opportunities* afforded by *Continuous Improvement* practices which include exploring innovations, improved integration through multisectoral approaches toward nutrition, agriculture, microfinance and SBCC programming with SPPs in both country settings. Additionally, monitoring and evaluation was cited by stakeholders with research expertise as an area which could demonstrate to government and donors the potential for SBCC to achieve improved behavioural impact. Other *Opportunities* identified included schools' programs, nutrition microfinance or 'kitchen gardens' agribusiness incentive opportunities (see Table 2. Opportunities for Program Improvement).

Given the importance of understanding current awareness and attitudes toward SPPs and nutrition related support programs, an important thematic investigation related to beneficiaries *Knowledge, Attitudes, and Practices (KAP)*. KAP

**Table 2. Organising Theme – Opportunities for Program Improvement.**

| Sub-Theme | Feedback |
|---|---|
| Integration | "We give seeds and fertilizer in 3 seasons to our marginal farmers who have at least 1bigha land. Generally, 2000-3000 farmers get the facility. We give them input; we don't give them any money directly. And those who don't have any land we suggest they cultivate in sacks, but we don't give them any input, we just give them suggestions. We arrange some yard meetings and training as well. There we give them nutrition related messages as now our government is emphasizing nutrition greatly. Here, we have almost 12,000 farmers engaged."<br>*Rural Front line Field Worker/SPP Service Provider, Kurigram, Bangladesh.*<br>"If you can bring together the nutrition component, health component, gender component and disability component then it will be more beneficial. If they increase the inter-ministerial communication and make coordination strong then I think making the safety net stronger would not be so difficult."<br>*Director General, Government Nutrition Agency, Bangladesh.* |
| Capacity Building/ Empowerment | "During the Imam training, training is given on nutrition, food production and they are also rewarded for good performance in food production. Imams are encouraged to talk about nutrition in their sermons. Imams are also given interest-free loans and one-time assistance to the poor."<br>*Urban Front line Field Worker/SPP Service Provider, Patuakhali, Bangladesh.*<br>"Whatever program, whether it is social protection, there should be a pull strategy in that community rights and entitlements are emphasized and help to instill a sense of ownership within the community, fostering the belief that these programs are theirs and that they have a rightful claim to certain benefits."<br>*Medical Marketing and BCC Specialist, INGO, Islamabad, Pakistan.* |
| Co-Design of Messages | "Information should be provided in a local Service Centre: Info to include: Who will get the SPP benefits? Which documents are needed? When will the benefit come?/When will people get it?, Where can we complain?<br>*Rural-Females-19–34 Years-CD-SES, Co-Design Workshop, Kurigram, Bangladesh.*<br>"The Lady Health Worker program in Pakistan, or the Midwifery program; they are connected to the community. You teach them, engage them, train them. But they don't have to give lectures. You have to demonstrate the methods. We still don't know what methodology will ensure a reduction in malnutrition."<br>*Visiting Faculty Aga Khan University, Advisor Community Engagement, Pakistan.* |
| Community Engagement | "We need to think in a very focused way. So that no one will go out of the safety net, and we can seal the gaps/leakage of the net. So, we need more community representatives, by groups, by villages/para. It can be one woman for 10 women, she might be not the elected representative, but she represents the community, such as community worker, community leaders those who are interested."<br>*Director General, Government Nutrition Agency, Bangladesh.* |
| Continuous Improvement | "I guess we could build an agriculture club including all vulnerable people. We have to develop them."<br>*Rural Front line Field Worker/SPP Service Provider, Kurigram, Bangladesh.*<br>"I was thinking that counseling is in the BISP's national program. But we have never evaluated the outcome. You need to evaluate it so that your next strategy is better. So, you need to work on this."<br>*Senior Advocacy Advisor, Chairperson INGO SPP, Pakistan.* |

sub-themes of program *Awareness/Knowledge* identified challenges on where SPPs were located, what services are provided, what documentation is required to apply, and how applications need to be processed. Similar concerns about poor knowledge on nutrition were demonstrated including understanding of what types of food to eat, meal frequency, the need for dietary diversity, and what specific infant and young child feeding practices should be followed.

The second *KAP* organizing theme, sub-theme—*Attitudes and Perceptions* included investigation on important behavioural determinants of self-efficacy perceptions [23] and perceived behavioural control [24] emanating from the behavioural literature. While stakeholders working to improve nutrition outcomes have emphasised the importance of greater integration of nutrition-sensitive SBCC interventions with SPPs, the feedback from a number of beneficiaries of SPPs highlights their preference for cash transfers with the money received providing discretionary spending for the limited funds provided. However, other feedback from stakeholders indicates that cash transfers may often be spent by male, heads of households, on items that may not benefit women and children's nutritional needs.

Other feedback from stakeholders emphasises that *Attitudes* which lead to *Intentions* and *Behavioural Change* are likely more important determinants to shift people along the behaviour change continuum, than simply awareness or knowledge on their own.

The *Social and Cultural Norms,* originally emanating from constructs of "Subjective Norms", identified as "*a function of the normative beliefs of a society and the motivation for someone to comply with each important person in someone's life*" [25] emerged as an important behavioural determinant. Feedback from stakeholders and beneficiaries emphasises the social and cultural barriers that impede vulnerable groups –mostly women– to achieve behavioural control. *Motivation* [26], which can lead to *Social Mobilization,* was another important sub-theme of KAP determinants that can be more fully impacted through greater participation by community influencers, with the subsequent skills and confidence that can evolve from participant engagement in SBCC including the reshaping of negative social and cultural norms.

The final sub-theme relating to KAP is *Intentions and Behaviours,* which includes *Trialling Behaviours.* The range of behavioural determinants highlights the often-long journey to move individuals and communities from *Awareness* of specific programs to *Intentions* to engage in activities, *Trial* the recommended behaviours, and finally *Adopt* and become *Advocates* for the behaviours (see Table 3. Knowledge Attitudes and Practices [KAP] Indicators).

The last area emerging from the thematic analysis under the organising themes of *Target Groups*, *Messaging* and *SBCC Entry-Points* highlights a number of sub-themes emerging, which are key to the development of an evidence-based *Theory of Change* on SBCC in nutrition-sensitive SPPs. The first sub-theme in this area relates to target group selection and the fact that both *Demand* and *Supply-Side* groups will need to be considered. A number of priority target groups were highlighted including targeting to those already engaged with SPPs, as well as those currently not on the programs. Second, is the need to prioritise particularly vulnerable groups including pregnant, lactating or non-lactating mothers, and vulnerable youth. Next, is the need to also target –as a lower priority– supply-side groups including key influencers from community organizations, SPP staff, community health center staff, NGO staff, and voluntary health workers, including lady health workers prevalent in Pakistan. Other important influencers could include community leaders, religious leaders and teachers.

As a follow-up, feedback on the organizing theme of *Message Types* highlights the importance of considering the sub-theme of *Social and Cultural Norms* and how they influence community beliefs, traditions, and dietary preferences. Next, was the organising theme to identify *SBCC Entry-Points,* including sub-themes of *interpersonal communication* (IPC) sources which were overwhelmingly preferred to deliver dietary advice and SBCC materials by vulnerable community members and many stakeholders in both country settings. This approach is seen to provide more personalized communication from local influencers, including community leaders, and other respected figures who are seen as highly credible message sources within their respective communities. As such, these individuals, if well trained on SBCC approaches and provided appropriate support materials, would likely be in the best position to address specific community concerns on SPPs and nutrition-sensitive SBCC. IPC approaches identified by some stakeholders and beneficiaries included door-to-door visits from influencers with the acknowledgement that this approach may not be economically viable at-scale. As such, greater integration could be considered in utilizing existing human resources for IPC, including voluntary health workers already operating in most village settings.

The next most popular sub-theme for the *SBCC Entry-Points* organising theme, popular in both countries was *Mobile Messaging and Social Media*, with the mobile application 'WhatsApp' being frequently seen as the predominant communication modality for health and other messages in rural areas of Pakistan. Given the relatively lower literacy rates in Pakistan, it was recommended that recorded oral messaging be used in place of text messaging, where-ever possible.

Another sub-theme of the *Entry-Points* organising theme, highlighted by a smaller number of respondents in the co-design sessions was *Print Media* and *Community Media*, acknowledging that some print community resources might require improvements to the design of the resources through message pre-testing. Print resources were also highlighted by supply-side stakeholders given their potential to support IPC delivery of purposive messages by opinion leaders in their communities. Next, was the sub-theme of *Mass Media*, which was often mentioned by both stakeholders and beneficiaries due to its national proliferation. Many other health issues were recalled via televised media while the considerable cost of purposive messaging on national TV channels was highlighted by country media communications specialists. Community and local radio stations were also occasionally mentioned while community 'miking' —recorded message played through

**Table 3. Organising Theme - Knowledge Attitudes and Practices (KAP) Indicators.**

| Sub-Theme | Feedback |
|---|---|
| Awareness/Knowledge | "There is lack of awareness. People don't know where offices are. What documents will be required. How to apply. And lack of trust. People also don't want their females to go to offices for enrolment."<br>*Adult-Rural-Male-C-D SES, Sheikhupura Village, Pakistan.*<br>"One of the major causes of child stunting is that mothers do not get enough food during pregnancy. Not only because they are poor, that is the major reason, but also, they do not have the knowledge about what type of food to eat, when to eat, what kind of infant and young child feeding practices should be followed. So, nutrition BCC design addresses these issues."<br>*Country Representative, INGO Food Policy Research Agency, Bangladesh.*<br>"I would say awareness gaps are persistent at the grassroots level for SPPs and hence those who are genuinely needy cannot access it. This indicates a need for targeted efforts to reach and educate vulnerable populations directly."<br>*Director Public Relations, Government Social Security Agency, Pakistan.* |
| Attitudes/Perceptions | "We like that BISP comes in cash form. We can use it as per our needs. We want support programs only in money form."<br>*Adult-Rural-Female-C-D SES, Sheikhupura Village, Pakistan.*<br>"The knowledge is good I think, but we must work to change their attitudes."<br>*Ex-Line Director, Government Nutrition Services, Bangladesh.* |
| Social/Cultural Norms | "In the context of a child experiencing diarrhea, the recommended solution is the administration of Oral Rehydration Solution or guidance. In these, societal norms plays an importance in defining the roles and responsibilities of women. In our society a good woman is quiet; she listens to the husband, doesn't answer him, gives him food, does everything."<br>*Visiting Faculty Aga Khan University, Advisor Community Engagement, Pakistan.* |
| Motivation/Mobilisation | "I like to work with this grass roots level people. But they are really naive. But when we can't motivate our farmers then I feel bad. They keep asking, 'what we'll get? How will we be benefitted?' We couldn't motivate them actually."<br>*Front line Field Worker/SPP Service Provider, Kurigram, Bangladesh.*<br>"The poor are being ground everywhere, like flour is ground in a flour mill. A poor person; whether it's access to government hospitals, education, necessities like food, or any other aspect of existence, the poor endure while the rich continue to prosper."<br>*Adult-Rural-Female-C-D SES, Sheikhupura Village, Pakistan.* |
| Intentions/Behaviours | "We have to emphasize the importance of behavioural change, urging not only a shift in behaviours but also a broader commitment to responsible and informed decision-making within vulnerable communities for the amount of money they get from SPPs."<br>*Head of Office & Lead for INGO Food Program, Pakistan.* |

loudspeakers on bicycles— was also mentioned in the Bangladesh context, given the long history of its use in rural village settings. Important to emphasise in terms of SBCC message *Entry-Points* was the general desire to include priority messages at every contact point, with expert advisors emphasising that messages needed to be workshopped and pre-tested with the communities most impacted by the program.

The last theme to emerge from the analysis was the issue of *Incentives or NUDGES* [27], with the sub-themes of *Conditional Cash Transfers* (CCTs) and the potential for improved food supplementation highlighted by a number of SPP stakeholders. Several existing programs in Bangladesh and Pakistan already appeared to be using food or cash incentives highlighting the potential to *Nudge* mothers with children to visit health facilities and access IPC health worker channels during their visits. Additionally, CCT NUDGES were also used to build nutrition knowledge with program beneficiaries through targeted community training sessions which were required as part of the CCT benefit (see Table 4. Target Groups, Messaging and SBCC Entry-Points).

## Discussion

This formative study integrating co-design has shed light on some of the key entry-points for nutrition SBCC interventions in SPP identified by the program beneficiaries. This includes the types of messages, advocacy approaches and

**Table 4. Organising Themes- Target Groups, Messaging and SBCC Entry-Points.**

| Organising Theme | Sub-Theme | Feedback |
|---|---|---|
| Targeting | Demand and Supply Side Groups | "The packages should be both for those with the safety net, and those without the safety-net. There should be three packages, for all adolescents, pregnant, lactating or non-lactating mothers. I want to see messages for the safety-net for all urban-rural groups, the 'have'- and 'have-not' groups." <br> *Ex-Line Director, National Nutrition Services (NNS), MOHFW Bangladesh.* <br> "Similarly, community-based organizations are directly connected with vulnerable people. They can be trained quickly because they are present in the community. They listen to people. Thirdly, community health is very important for training so that they can focus properly." <br> *Senior Advocacy Advisor, Asia. Chairperson Global SUN Civil Society Network, Pakistan.* |
| Message Types | Social and Cultural Context | "Nutrition messages should link with the cultural context of the target audience. Local beliefs, traditions, and dietary preferences are more relatable to the vulnerable population. We can also use community engagement sessions where nutrition experts can interact directly with community members. This allows for a more personalized approach, addressing specific concerns and answering questions. Partnership with local influencers, community leaders, and respected figures to communicate nutrition messages." <br> *Medical Marketing Specialist – Pakistan Maternal Nutrition Campaign & BCC Specialist, Nutrition International, Islamabad.* |
| Entry-Points | Interpersonal Personal Communication | "The best, if most expensive, method is IPC (interpersonal personal communication), which allows us to visit households and deliver dietary advice and SBCC materials orally." <br> *Deputy General Program Manager, Bureau of Health Education* <br> "The entry points for these SPPs might be in the rural base; those who participated in the implementation and distribution by a committee could potentially serve as the entry points." <br> *Research Director (Nutrition), Food Planning & Monitoring Unit, Ministry of Food (MOF), Bangladesh.* |
|  | Mobile Messaging/Social Media | "Social media is good as people have mobile phones. We only have this mobile phone, no other source to find information." <br> *Adult-Urban-Male-C-D SES, Johar Town, Lahore, Pakistan.* <br> "The majority of us go through the prior and create cut-and-paste formulas while preparing for the IEC 'martial arts'. As you can see, not just us but also child groups, teenagers and the elderly have taste changes from time to time. As a result, they are unable to consistently perceive the same messages from the same counselling. Rather with the three, four, or even five-minute recordings we used to make, we now make 30-second Tik-Tok films." <br> *Program Manager, National Nutrition Agency, Bangladesh.* <br> "You can use social media to share recipes, nutrition tips, and stories of people who have improved their nutrition. The key influencers, like many people for Polio, used the Maulvis." <br> *Senior Advocacy Advisor, Asia. Chairperson Global SUN Civil Society Network, Pakistan.* <br> "They should send messages on WhatsApp or send simple phone messages. Telling us about the eligibility in the program." <br> *Adult-Rural-Female-C-D SES, Sheikhupura Village, Pakistan.* |
|  | Print Media/ Community Media | "They (Gov.) are still using old style brochures, handouts and posters which are hardly revised with no innovation added to its content. Even the advertisements of many of our public health and social safety programs are not reaching those who need them because the way they are being advertised. We need to work very hard to come up with innovative ideas. Our youth needs to be engaged. They should be used as champions of change." <br> *Director Public Relations, Social Security Punjab, Punjab Social Welfare Department, Pakistan.* <br> "I am interested in interpersonal communication, but we do not have enough human resources, so we must go for print or mass media." <br> *Ex-Line Director, National Nutrition Agency, Bangladesh.* |
|  | Mass Media | For vaccination we see messages on TV and teams also make announcements through Mosques and they also visit door to door. We trust them all." <br> *Adult-Rural-Male-C-D SES, Sheikhupura Village, Pakistan.* <br> "We must focus on changing behaviors. There is a need for a more targeted and culturally sensitive messaging strategy for SPPs to effectively communicate with the intended beneficiaries. Every house has a TV. Even every poor houses have a TV. They also have mobile phones." <br> *CEO, National SPP, Pakistan.* |
| Incentives/ NUDGES | Conditional cash transfers/Food supplementation | "However, this additional cash in the 'conditional cash transfer' is specifically allocated for the purpose of encouraging mothers and children to visit health facilities. They can collect both cash and food during these visits." <br> *Program Officer, Nutrition, Integrated Reproductive, Maternal, Newborn & Child Health (IRMNCH) Nutrition Program, Govt. of Punjab, Pakistan.* <br> "What we did in our program is we made cash the transfers in the case of food, conditional on attending this nutrition BCC. So, you attend this training every week, then next month you get the transfer. If you don't attend, you don't get it. So, this conditionality also worked very well." <br> *Country Representative, INGO Food Policy Research Agency, Bangladesh.* |

dissemination channels most applicable for use in Bangladesh and Pakistan. The qualitative approaches to the needs assessment and more specific probing on barriers and challenges identified by participants during the co-design sessions enabled more in-depth investigation to achieve culturally appropriate SBCC solutions to support the findings of the systematic review [15].

A synthesis of the findings across groups identified a number of important *KAP indicators:* Poor knowledge about nutrition as well as how to enrol for the schemes, and poor attitudes and self-efficacy perceptions emanating from participants inability to successfully enrol or qualify for SPPs and achieve nutritional benefits offered. This is in line with other studies which have identified that households of marginalised groups may lack sufficient information and education to understand the value of early life nutrition [28,29]. Knowledge about the correct foods to eat and the importance of dietary diversity for better health was more apparent with Bangladeshi respondents than Pakistani beneficiaries. This possibly reflects the smaller geographic footprint of Bangladesh, and the proliferation of media and community information sources coupled with the greater efficiency in disseminating messages through mass media available in the country [30] vs Pakistan's more expansive and remote, rural regions and the challenges in reaching remote tribal and religiously conservative populations.

"I got nutrition information from a doctor at a health complex, from community drama, from yard meetings. They delivered various nutrition related messages like how the baby will get nutrition, that we should eat milk, eggs, vegetables, fruits, and then the baby will be healthy: We got it from an NGO worker, through miking, through mobile phones, TV, News. Community clinics also provide these nutrition messages."

*Rural-Females-19-34 Years-CD-SES, Kurigram Co-Design Workshop, Bangladesh.*

The lack of [23] *perceived behavioural control* by women, particularly in rural communities in Pakistan suggests the role of SBCC may be to subtly shift perceptions of social and cultural norms in strict conservative communities, to allow for greater agency by those most involved with the health of children and other family members. With poor *self-efficacy* [23] and perceived behavioural control comes poor motivation and social mobilization of most vulnerable groups. The findings indicate the need to consider more innovative ways to engage poorly motivated groups through incentives and other promotional benefits.

Five *barriers to program engagement* emerged from the analysis including resource constraints, corruption/nepotism in the administration of the programs, stigma and discrimination felt by a number of vulnerable applicants to the programs, and the role of gatekeepers -predominantly husbands of women- who prevented women's access to the SPPs. The lack of strategic approaches were also seen as a major barrier to the successful administration and promotion of the program through SBCC.

The role of gatekeepers featured more so in Pakistan than in Bangladesh with the restrictive role of male gatekeepers emphasized in relation to family food decisions, with FGD moderators in Pakistan reporting that husbands and other family members also attended the FGDs with women respondents and had to be ushered into other areas while the groups were being conducted. This highlights current cultural and social norms which inhibit women's agency to seek SPP and nutrition support, also reported from studies in other religiously conservative rural settings; alongside recommendations to also target men in these settings with nutrition-sensitive SBCC program messaging [31].

Issues of stigma and discrimination also need to be addressed with SPP providers given the competitiveness of people trying to access the schemes and the humiliation experienced by those who were rejected in favour of participants perceived to have familial ties or relationships with the administrators of the schemes. Along with the perceived poor administration of some SPPs, were often cited barriers to enrolment including the complex documentary requirements and online application procedures which many vulnerable respondents were not able to negotiate.

Furthermore, the co-design process conducted with program beneficiaries revealed that, across both countries, there were negative opinions of SPPs, funding levels to support adequate food supplementation, and other household

expenses. This was coupled with the shame and humiliation reported by some applicants, and other poor customer experiences that in some cases, may have been designed to discourage engagement. The study findings confirm other studies of the limited coverage of SPPs, including weak and inadequate cash transfer coverage to poor women's groups; the reinforcement of a cycle of privilege and disadvantage among socioeconomic classes, errors in distribution of existing schemes, and challenges faced by non-beneficiaries in gaining benefits [32–34].

With the challenges, five opportunities for program improvement were also identified including opportunities for improved integration of SPPs with nutrition programs and SBCC to promote the programs and address beneficiary concerns. Capacity building and empowerment of vulnerable groups, particularly women with young children, was identified as an important requirement, as well as opportunities for co-design of messages to facilitate greater community engagement and address social and cultural norms within a culture of continuous improvement. Program donors have highlighted the ability of well-designed SPPs to reduce women and girls vulnerability, improve nutrition, enhance education outcomes, protect livelihoods and facilitate women's economic and social empowerment [35].

Discussion on the organising theme of knowledge, attitudes and practices (KAP) indicators, identifies the need to focus on key *behavioural determinants* to support and maintain behavioural change. This includes building awareness and knowledge about SPPs and access to services, changing attitudes and perceptions, both from the supply and demand sides, toward the optimal functioning of the programs, addressing social and cultural norms in messaging and promotional activities to motivate and mobilise populations toward greater behavioural intentions to get enrolled in the programs and improve nutrition outcomes.

This emphasises the importance for SBCC interventions that focus on achieving behavioural change among the most vulnerable, while also ensuring SBCC activities will move other participants at various stages of change, further along the *behaviour change continuum* [36]. This includes the need for pretested messages delivered through highly credible IPC message sources of key influencers and opinion leaders which have also been identified as key drivers for change in other development programs [37].

Possible solutions provided by respondents in both countries from the co-design sessions included the need to establish an independent complaints mechanism in the form of an Infoline, easily accessible by community members. This accords with recommendations from other studies for more "information-related channels" for nutrition and social protection available to the vulnerable [38,39]. Other feedback recommended more community awareness sessions be conducted, so community members were better informed about the nature of the SPPs, what benefits were provided, the level of the benefits, when the benefits would be received, and the assessment criteria for enrolment into the programs [39]. Barriers to improved nutrition within the communities of the poorest of the poor identified that the basic food packages provided with most food programs usually only addressed food hunger, rather than the provision of more nutrient-dense foods and adequate dietary diversity.

Opportunities for SBCC identified the greater need for program integration including CTTs linked to improved health outcomes, capacity building and empowerment of beneficiaries lacking in confidence and ability to effectively engage with the programs and resultant benefits to be accrued, and greater community engagement in the effective design and delivery of the programs including co-design opportunities at various stages of the program design, with the approach seen as most often used for nutrition intervention development [10].

The formative research and systematic review has highlighted the importance of addressing the barriers to improved nutrition through nutrition-sensitive SBCC approaches embedded within cash transfer programs. This could include building awareness and knowledge on local, culturally acceptable and affordable nutrient-dense foods, and the provision of child-appropriate recipes which were identified through the systematic review [40,6]. Opportunities to integrate nutrition with the conditional cash transfer programs also considered the role of NUDGES or incentives to qualify for the CCTs which were contingent on greater engagement by program beneficiaries to build knowledge and capacity for improved nutrition outcomes, particularly through trainings attended by women with young children. CCT NUDGES could also

include beneficiaries taking their children for regular health checkups at Primary Health Centres (PHCs) to monitor growth and development. Similar approaches in providing cash alongside nutrition counselling, have been found to enable vulnerable populations to adopt more positive nutrition practices, and improve the diets of young children [39].

Food stamps/vouchers was another idea raised by stakeholders in Pakistan that is worth trialling in both countries, as it allows recipients to use a portion of funds provided in the form of food vouchers to purchase specific nutritious products, rather than using the funds for other discretionary spending. The approach, which has been adopted in a number of high-income country settings, testifies to the efficacy in channelling limited safety-net resources toward improved nutrition and health outcomes of the most vulnerable. Despite the reported improvements in dietary outcomes, the system will need careful planning due to reported unintended effects, due to purchasing restrictions applied to some food voucher programs [41].

Issues related to the range of SPP providers with different program branding indicate that many beneficiaries, particularly in Pakistan, are not aware of the term "social protection program", despite many being enrolled in the programs. Additionally, the large number of SPPs operating in both countries (more than 100 SPPs identified in Bangladesh) are likely to create confusion among beneficiaries on where to enrol in the programs and what nutrition and cash benefits are provided. The politicisation of SPPs has also caused challenges in enrolment with these branding issues also highlighted in other country settings [42,43].

## Recommendations

The study has highlighted the need to address a number of structural and policy issues, if nutrition-sensitive SBCC interventions are to significantly contribute to improved health outcomes for SPP recipients in Bangladesh and Pakistan. A supply-side issue identified by stakeholders in both countries is the need for improved strategic approaches including the development of more comprehensive and integrated SBCC interventions.

Next, is the need to encourage policy development for greater integration of nutritious food provision with the cash transfer programs with the evidence from the needs assessment and the systematic review pointing to the added benefits of integrating cash transfers with nutrition SBCC [44,45].

> "They wanted to see the impact of providing only cash transfer, food supplement and cash transfer, and cash transfer plus supplement and nutrition BCC. The study has shown benefits to include all three arms. So, I foresee a great opportunity in adding a nutrition behavior change strategy into the SPPs."

*Head of Office & Lead for INGO SPP Food Program, Pakistan.*

However, there is currently a relatively small component of nutrition or food provision in SPPs currently operating in Bangladesh, and to a lesser extent, in Pakistan. This is despite evidence that confirms the importance of integrating nutrient supplementation with SBCC into SPPs in many LMICs [28,46,47]. Of the food supplementation operating in conjunction with SPPs in Bangladesh and Pakistan, it's also clear from stakeholder feedback that, other than for agri-focussed, kitchen gardens programs, the staple foods provided with SPPs supplementation –rice, oil, and dried pulses– are mainly provided to reduce hunger rather than improve nutritional outcomes with most vulnerable family members—women and their children. Therefore, policy to integrate conditional cash transfer programs with more nutritious food incentives promoted through SBCC may provide better nutrition outcomes for vulnerable beneficiaries.

> "They usually just provide rice to individuals in need; they do not, however, provide cashews or any other food that is high in nutrients, like fruits, vegetables, etc. Before it was terminated, this program was only expanded to two distinct regions. I found that although there were cash vouchers available for fruits and vegetables as part of the government initiative, most individuals were more interested in the fortified rice than in other nutrient-dense meals."

*Communication Specialist, Bangladesh Center for Communication Programme (BCCP)*

This recommendation is despite beneficiary preferences for cash benefits allowing them to prioritize spending for their families. The evidence from the needs assessment indicates that with those in high need of improved nutrition, cash transfers on their own, may often not be allocated to the provision of nutritious food, but rather be used for other discretionary spending. Additionally, to alleviate the resource constraints of food programs linked to SPPs, agricultural integration and other multisectoral approaches should also be considered in areas where food can be more easily cultivated, or microfinance opportunities can provide protein dense dietary diversity. These approaches also provide greater self-sufficiency and sustainability while building competencies, confidence, skills and empowerment of vulnerable groups, often stigmatized by the idea of government hand-outs [48].

"One thing I would like to change? Train farmers to encourage them to grow nutritious food in this area."

*Urban Front line Field Worker/SPP Service Provider, Bangladesh.*

Furthermore, integrating nutrition-sensitive SBCC interventions into SPPs can further enhance the sustainable impact of the programs in regard to food quality and security, dietary intake and diversity and are seen as imperative in fostering enabling environments, including kitchen gardens programs to ensure improved growth and nutrition for mothers and their children [49]. Coupled with evidence that cash transfers alone do not alter parenting practices or improve early childhood development and food security [50], greater efforts must be made to integrate these programs.

"Children admitted to our pre-primary schools will be paid Tk75 if they have 80% attendance -Tk150 per month for a child if they are going to school in our 1-5 classes. Two children in the family will get Tk250. Nutrient-rich biscuits were provided in sub-districts. This was under the World Food Programme. It would have been better if biscuits were given instead of money."

*Urban Front line Field Worker/SPP Service Provider, Bangladesh.*

## Identification of target groups

Recommendations are for the identification of primary, secondary and tertiary target groups, in line with national SBCC experts needs assessment advice, with differing SBCC objectives to meet each of the group's specific needs. Primary target groups for nutrition-sensitive SPPs should include women of childbearing age, pregnant women, women with babies 0–12 months, and women with children 1–5 years of age. This aligns with the identification of vulnerable groups across the life cycle by WFP and includes women and children in the first 1000 days, preschoolers, schoolchildren, and adolescents [51]. Other programmers have acknowledged that these programs aim to improve the health of most vulnerable population groups, including pregnant, lactating mothers and children, particularly in LMICs [52].

Secondary target groups should include general population groups of all adult females and males to build nutrition literacy in the broader population. Tertiary target groups, who may be the subject of SBCC capacity-building, include supply-side stakeholders and key influencers such as front-line fieldworkers, lady health workers (Pakistan), community health workers, religious leaders, teachers, local leaders, and government, INGO/NGO and civil society staff involved in SPPs and nutrition interventions. As noted, the specific behavioural objectives and SBCC message designs will need to vary for each target group, given their different needs and wants, with these processes determined through additional co-design strategic planning following dissemination and endorsement of the nutrition-sensitive SBCC Entry-Point Platform.

## SBCC message entry-points

A number of distinct entry points for SBCC nutrition messages to support SPPs were identified by stakeholders and program beneficiaries, with other studies also confirming the importance of targeted messages, delivered at critical times, to increase consumption of nutrient dense foods and improve food security [28]. What is evident from the study is the importance of initiating discussion on food insecurity and nutrition with highly vulnerable groups at the very first point of contact and at every SBCC entry point following the initial enquiry. Recommendation from co-design sessions highlighted the need to link messages to the social and cultural context of target audiences of vulnerable rural populations whose local beliefs, traditions, and dietary preferences will need to take precedence when promoting messages on sustainable nutrition behaviours. It's recommended that all of the SBCC entry-points highlighted by stakeholders and beneficiaries should be considered, with a particular focus on channels –IPC through key influencers, Mobile Phones, and Social Media– most often mentioned as primary channels of communication.

## SBCC message dissemination

SBCC message dissemination is an important component of the SBCC Entry-Point Platform with the recommendation that the most preferred and impactful messages should be delivered by highly credible interpersonal communication (IPC) sources. These comprise –key influencers and opinion leaders including doctors/nurses, and other Government and NGO health workers, religious leaders, local leaders, community members (MPs), and allied health and agricultural professionals. Other credible IPC sources identified were friends, family members, youth, teachers and children in schools. This recommendation conforms to other study recommendations for key messages to be delivered through community mobilization processes, whereby community or social workers, health workers, peer educators and counsellors are trained and engaged for health promotion and education [53].

Door-to-door dissemination of IPC messages was one approach recommended by many respondents, while stakeholders acknowledged the high costs involved with this form of message dissemination. Therefore, IPC delivered through community settings and purposive trainings would be more economical. Other preferred message dissemination methods included mobile media with recorded messages on WhatsApp (for illiterate community members) and community "miking" (Bangladesh only). Social media was also seen as a cost-effective and pervasive message source easily accessed by many people via their mobile phones, with Facebook and Tik Tok most frequently mentioned, alongside the use of local social influencers with large followings to front social media campaigns. Community radio/talkback radio, and community media such as posters and outdoor billboards should also be considered where budgets allow, with more costly purposive TV mass media messaging and news and current affairs programming considered when there is a strong buy-in to SPP and nutrition programs SBCC from government and donors.

Additionally, recommendations are to expand the application of *NUDGES* in the form of food incentives to empower community members and achieve a greater commitment to a range of health improvement behaviours:

> "So, we are providing specialized nutritious food to the beneficiaries because we believe that the poorest of the poor families are facing this food insecurity issue. So, that is why we incorporated this specialized nutritious food in our model. Then we are also ensuring their immunization in the program. Then antenatal and postnatal care is one of the components in the program, and then awareness sessions: The beneficiaries must attend the awareness sessions by visiting our health facilities."

> *Director General, of Government Cash Transfer Program, Pakistan.*

Next, is the call for improved strategic planning through a nationally endorsed nutrition-sensitive SBCC strategy and action plan which includes Monitoring, Evaluation, Learning and Adaptation (MELA) components to measure program inputs, outputs, outcomes and impact, and use lessons learned to continually improve on interventions. A

SBCC strategy endorsed by the government can also more clearly articulate key components required to achieve nutrition objectives and provide a practical stepwise approach to improve integration of nutrition-sensitive SBCC into SPPs. This would include more systematic, evidence-based activities for the design and pretesting of messages, with opportunities to scale-up messages which resonate with target groups and support nutrition behavioural change objectives.

> "In Tawana, the concept was to promote local agriculture; the community was so happy because they could relate better diets with local foods and dishes. We didn't talk about a 'nutritious diet', only 'power giving foods' with the pictures: They related to it very well. How do you translate it further?"

*Former Director of donor funded school lunch project, Pakistan.*

Last, is the importance of continuous program improvement which can only be achieved through well designed, nutrition-sensitive SBCC interventions which are effectively monitored for their impact with the view to scale-up approaches that are found to provide the greatest benefits and cost-efficiencies. As such, donors and governments should also be encouraged to provide adequate funds for MELA components of these programs to measure the effectiveness of SPPs on nutrition outcomes which require considerable investments over time. The recommendations emanating from the needs assessment and supporting systematic review have been articulated through the development of a SBCC Entry-Point Platform Spreadsheet and key components (see Table 5).

## Limitations

Limitations of the study is that in moderation of FGDs group dynamics and moderator influence—including unintentionally influencing participants' responses through moderator biases, body language, or leading questions—may result in skewed data that may not accurately reflect participants' true opinions. Other limitations include unequal participation or dominant participants in groups which may lead to the voices of most vulnerable participants being underrepresented in the discussions. Next, is the issue of maintaining group focus for study protocols which may take up to two hours to complete. The handling of sensitive topics around malnutrition and chronic diseases can also skew results, due to participants avoidance in discussing uncomfortable issues. Additionally, the complexity of the data, environmental influences, and the limited depth of discussions with all participants can also impact on the validity of the insights gathered. In some cases, the non-anonymity of focus group participants may inhibit individuals from sharing their true thoughts and feelings. Additional limitations of co-design approaches is that the process is likely more taxing to groups, given the range of stimuli being presented, and the complex number of issues covered. From a data analysis perspective grounded-theory approaches to qualitative research can also be heavily directed by a primary researcher, potentially leading to internal biases. Despite the limitations, the authors believe the study can still claim generalizability in relation to country findings given the adequate sample size and multiple data-sources from stakeholders and beneficiaries across both country settings. As such, the qualitative processes have also benefitted from substantiation through quantitative statistical measures analysed through the systematic review.

## Implications for future research

Given the potential for the integration of co-design into formative research stages of priority health programs, to provide more culturally specific solutions to SBCC program design, future studies should look at creative ways to include the beneficiaries of health programs into the design of the programs. This could include more opportunities for beneficiaries to contribute to the development of socially and culturally appropriate messages, creative approaches as well as providing advice on the most efficient dissemination channels for SBC messages and solutions given their lived experience of the problems that programmers are trying to address.

 

**Table 5. Nutrition-Sensitive SBCC Entry-Point Platform Spreadsheet.**

| | Priority Target Groups | Key Messages* | Dissemination Channels ⱡ | | | | | |
|---|---|---|---|---|---|---|---|---|
| | | *Raising Awareness, Building Risk Perceptions, Self-Efficacy Perceptions, Changing Attitudes, Increasing Capability, Motivations, Intentions, Encouraging Trialling, Adoption, Maintenance and Advocacy for recommended preventative Behaviours.* | IPC[1] | Mobile Phones[2] | Social Media[3] | Radio Media[4] | TV Media[5] | Print/ Community Materials[6] |
| **Primary Target Groups** | 1. Women of Reproductive Age  2. Pregnant Women  3. Women with babies 0–6 mths  4. Women with babies 6 mths–2 yrs  5. Women with babies 2 yrs–5yrs. | • Early initiation of breastfeeding within 1 hour of birth.  • Exclusive breastfeeding for the first 6 months of life.  • Introduction of nutritionally-adequate and safe complementary (solid) foods at 6 months together with continued breastfeeding up to 2 years of age or beyond.  • Identification of culturally appropriate local foods/recipes to meet nutrient needs and ensure dietary diversity with children and other family members.  • Seek out clinical support in difficult circumstances where special attention is required.  • Seek out SPP support if vulnerabilities are apparent.  • Website locations and an Infoline for nutrition information is provided. | ***** | **** | *** | ** | ** | ** |
| **Secondary Target Groups** | 6. General Adult Population. | • Malnutrition is a big problem in the country.  • SPPs provide income and food support to those in need.  • Application criteria is available at this online link.  • An Infoline is provided for concerns and complaints.  • Nutrition is a key aspect of SPPs to protect families from malnutrition.  • Identification of national nutrient dense foods/recipes for improved nutrition and dietary diversity.  • Avoid snack foods and highly processed foods high in fat, sugar and salt.  • Website locations and an Infoline for nutrition information is provided. | *** | *** | ***** | ***** | ***** | *** |
| | 7. Adolescents/ In and out of school children. | • Malnutrition is a big problem, affecting the young.  • Identification of nutrient dense foods/recipes and dietary diversity for improved child nutrition.  • Avoid snack foods and highly processed foods high in fat, sugar and salt.  • SPPs provide income and food support to those in need.  • Application criteria is available at this online link.  • Children should inform parents and other family members of support available. | *** | ***** | ***** | *** | ** | ** |
| | 8. SPP Managers and SPP front line field workers. | • Malnutrition is a big problem in the country.  • Greater integration of nutrition programs with SPPs is recommended to reduce vulnerabilities.  • You have a responsibility to inform vulnerable groups of food supplementation programs available with cash transfers.  • Cash transfer should be conditional on participants attending training and capacity building sessions and visiting community health clinics to get child health checks.  • Refer vulnerable groups to SPPs in your area and assist them to complete the required applications.  • Provide community members with website locations to apply and to Infoline's if they require additional support.  • Training packages are available from these sources. | **** | **** | ** | * | * | ***** |

*(Continued)*

| | Priority Target Groups | Key Messages* | Dissemination Channels ℓ | | | | | |
|---|---|---|---|---|---|---|---|---|
| | | *Raising Awareness, Building Risk Perceptions, Self-Efficacy Perceptions, Changing Attitudes, Increasing Capability, Motivations, Intentions, Encouraging Trialling, Adoption, Maintenance and Advocacy for recommended preventative Behaviours.* | IPC[1] | Mobile Phones[2] | Social Media[3] | Radio Media[4] | TV Media[5] | Print/ Community Materials[6] |
| **Tertiary Target Groups[X]** | 9. Opinion Leaders: Community Health Workers, Doctors, Local Leaders, Gov., INGO/NGO/ Civil Society staff involved in SPPs, Nutrition and Agriculture interventions. | • We all have a responsibility to address malnutrition and vulnerability in our communities.<br>• Identification of culturally appropriate nutrient dense foods/recipes for improved nutrition and dietary diversity to be shared with your community.<br>• Identification of SPPs in your area and the services they provide.<br>• Support vulnerable groups to apply and access cash transfer and food programs where required.<br>• Referrals of high-risk groups to local health centres and health workers – locations provided.<br>• Support materials for opinion leader advocacy are available on these websites and through the infoline. | *** | **** | **** | * | * | ***** |
| | 10. Local Health Workers: Front line fieldworkers, Lady Health Workers, | • Malnutrition is a priority issue for your government.<br>• Health workers who have high credibility in their communities have the greatest responsibility to address this issue with vulnerable groups.<br>• Identification of SPPs in your area and the services they provide.<br>• Support vulnerable groups to apply and access cash transfer and food programs where required.<br>• Provide health and nutrition information at every contact point.<br>• Support materials for local health workers are available on these websites and through the infoline. | **** | **** | **** | * | * | ***** |
| | 11. Religious Leaders, Teachers. | • Malnutrition is a priority issue for your government.<br>• Opinion leaders like Religious leaders and Teachers have a special responsibility to address malnutrition and other vulnerabilities with their constituents.<br>• Specialized and culturally appropriate nutrition packages and syllabus materials have been developed with your organisation's leaders.<br>• Training opportunities, resource packages and syllabus materials are available at these websites or through the Infoline.<br>• Ensure those trained spread the message to other community members | **** | **** | ** | * | * | ***** |

*(Continued)*

| | Priority Target Groups | Key Messages* | Dissemination Channels ᶩ | | | | | |
|---|---|---|---|---|---|---|---|---|
| | | *Raising Awareness, Building Risk Perceptions, Self-Efficacy Perceptions, Changing Attitudes, Increasing Capability, Motivations, Intentions, Encouraging Trialling, Adoption, Maintenance and Advocacy for recommended preventative Behaviours.* | IPC[1] | Mobile Phones[2] | Social Media[3] | Radio Media[4] | TV Media[5] | Print/ Community Materials[6] |
| **Key Actors to Deliver SBCC Messages by Country Setting** | | | | | | | | |
| **Targeted Interventions to High-Risk Groups in Community Settings** | | | | | | | | |

| BANGLADESH | PAKISTAN |
|---|---|
| 1. **NGO staff:** Working in SP and nutrition programs within local communities: Training workshops to synthesise messages and develop best practice resource kits to deliver key messages to community members.<br>2. **Community Health Workers/Nurses and Doctors:** Working within PHCs and other health settings: Training sessions and ToT to be conducted and key resource provided.<br>3. **Agricultural Officers:** Particularly those already working on field programs and conducting training with local communities. Specific resource materials like flip charts and posters to be provided to integrate nutrition-sensitive SBCC into their programs.<br>4. **Local Leaders:** Given their broad range of issues, simple key talking points and SMS messages to local constituents to be provided.<br>5. **Peer Educators:** Selection of local community members for training of trainers (ToT) and delivery of messages via IPC approaches in community settings.<br>6. **Local MPs and Government officials:** Simple talking points to be provided.<br>7. **Teachers:** Syllabus materials on nutrition to be developed or refined and training provided.<br>8. **Celebrities/ Role Models/ Social Influencers:** Engaged strategically and provided with specific training and SBCC resources as required. | 1. **Lady Health Workers:** Training and resources incentives to be provided given LHWs coverage of a range of other health issues in their communities.<br>2. **Imam's/Mullah's/Sheiks/Madrassa's** (particularly in conservative community settings): Specific training packages should be provided on key talking points.<br>3. **Agricultural Officers:** Particularly those already working on field programs and conducting training with local communities. Specific resource materials like flip charts and posters to be provided to integrate nutrition-sensitive SBCC into their programs.<br>4. **Local Leaders:** Given their broad range of issues, simple key talking points and SMS messages to local constituents to be provided.<br>5. **Peer Educators:** Selection of local community members for training of trainers (ToT) and delivery of messages via IPC approaches in community settings.<br>6. **Community Health Workers/Nurses and Doctors:** Working within PHCs and other health settings: Training sessions and ToT to be conducted and key resource provided.<br>7. **Local MPs and Government officials:** Simple talking points to be provided.<br>8. **Celebrities/ Role Models/ Social Influencers:** Engaged strategically and provided with specific training and SBCC resources as required. |
| **Key Agencies operating in each country setting Supporting SPP and Improved Nutrition Outcomes** | |
| **BANGLADESH** | **PAKISTAN** |
| • The Food Friendly Program – Khaddo Bandhob Karmasuchi – Health and Humanitarian Crisis Management Program, BRAC.<br>• International Food Policy Research Institute.<br>• National Nutrition Services, Ministry of Health and Family Welfare (MOHFW)<br>• Center for NCDs and Nutrition, BRAC James P Grant School of Public Health, Bangladesh.<br>• Food Planning & Monitoring Unit, Ministry of Food (MOF), Bangladesh.<br>• Bangladesh National Nutrition Council (BNNC), MOHFW.<br>• Global Alliance on Improved Nutrition (GAIN Bangladesh). | • National Socio-Economic Registry/Conditional Cash Transfer (Education, Health and Nutrition Benazir Income Support Programme), Government of Pakistan<br>• Punjab Nashonuma Program: World Food<br>Allah Walay Trust SPP, Pakistan.<br>• Pakistan Maternal Nutrition Campaign & BCC Specialist, Nutrition International,<br>• Global SUN Civil Society Network.<br>• Aga Khan University, Advisor Community Engagement, Indus Research Department & Ziauddin University.<br>• Nutrition, Integrated Reproductive, Maternal, Newborn & Child Health (IRMNCH) Nutrition Program, Govt. of Punjab, Pakistan.<br>• TAWANA Govt. School Lunch Project, Pakistan.<br>• Global Alliance on Improved Nutrition (GAIN Pakistan). |

*(Continued)*

**Table 5.** (Continued)

<table>
<tr><td>

**SPREADSHEET KEY**

*__Key Messages:__ Final messages should be resolved through participatory co-design/co-creation approaches with stakeholders and program beneficiaries to ensure social and cultural appropriateness. Messages developed should also be pre-tested with relevant priority target groups. Message layering with other community issues should also be considered.

X **Tertiary Target Groups**: Are the main message sources for Primary Target Groups and can also provide supplementary messaging to Secondary Target Groups. Identification of these IPC sources and key influencers as a target group acknowledges the need to 'target' these groups with SBCC training and resourcing and provide additional incentives for them to engage in the program. Important to note is that differences in predominant IPC sources have emerged from the country analysis with priority actors and key agencies identified in the supplementary table

ι **Dissemination Channels:** All dissemination channels should be considered where resource allocation permits. Otherwise, most cost-effective, largest reach and most impactful dissemination channels should be considered. Star rankings: *=Low Priority, **=Moderate/Low Priority, ***=Moderate Priority, ****=Moderate/High Priority, *****=High Priority.

1. **IPC channels** include: Opinion Leaders: Doctors, other Government and NGO Health Workers, Religious Leaders, Local Leaders, Community Members, Allied Health, Voluntary Health Workers and Agricultural Workers.

2. **Mobile Phone** priority Apps include: WhatsApp recorded messages; SMS messages should be considered with literate groups.

3. **Social Media** sites most often mentioned were Facebook and Tik-Tok with other popular sites considered based on national uptake data. Specific websites from SPPs/Nutrition and Agriculture agencies should also be promoted

4. **Radio Media** should consider radio talkback with opinion leaders over purposive campaign messages given the free cost of talkback radio. Community radio should be considered in rural/remote rural areas.

5.**Television Media** should consider news and current affairs programs featuring program champions and other opinion leaders. Purposive TV campaigns should be considered where funding is available. Negotiations should be made beforehand for additional free to air, bonus spots.

6. **Print and Community Media** includes community posters billboards, Community Miking (Bangladesh), and resource materials such as flip-charts, audiovisual materials used for training and capacity building. Pamphlets may also be considered but should be targeted toward opinion leaders and health workers to use as triggers for IPC.

</td></tr>
</table>

## Conclusion

The integration of co-design into qualitative methods to supplement a systematic review of the literature has provided important program intelligence from which to design culturally appropriate, nutrition-sensitive SBCC interventions to support SPPs in Bangladesh and Pakistan. The iterative approach to program design has also provided lessons learned for other LMICs addressing malnutrition. However, integration of co-design into initial stages of formative research should not be seen as a substitute for well-designed needs assessments that explore participant barriers and benefits to effective program implementation. Rather, the approach should complement all stages of formative research by allowing participants to engage more purposively into identifying possible solutions to the key issues raised through the qualitative enquiry. The inclusion of co-design sessions into initial stages of program formative research should also not preclude more comprehensive co-design sessions taking place with program beneficiaries at evaluative research stages and pre-testing following development of SBC communication resources and approaches. This iterative process of learning can be seen to provide greater opportunities for culturally nuanced and more readily accepted behaviour change programs to improve nutrition with the poorest of the poor. In particular, the integrated approach to formative research was found to be efficient in the resource-constrained settings of the study countries with the methodology evolving from needs assessments conducted for SBCC strategies for a range of other priority health programs, in LMIC settings [54–56].

Lessons learned indicate that the consultative approach of needs assessment formative research integrated with co-design sessions can provide greater ownership, participation and engagement by program beneficiaries as well as identifying more pragmatic solutions to the current barriers to entry to nutrition-sensitive SPPs, identified by beneficiaries.

## Supporting information

**S1 File. COREQ Checklist for Bangladesh Study.**
(PDF)

**S2 File. COREQ Checklist for Bangladesh Study.**
(PDF)

## Author contributions

**Conceptualization:** Tahir Turk, Anthony Wenndt.

**Data curation:** Tahir Turk.

**Formal analysis:** Tahir Turk, Rina Paul, Nilofer Fatimi Safdar.

**Funding acquisition:** Anthony Wenndt, Hamidullah Khan Babar, Tannaza Sadaf, Moniruzzaman Bipul.

**Investigation:** Tahir Turk, Rina Paul, Nilofer Fatimi Safdar, Nadia Shah, Syed Mahbubul Alam, Sohel Reza Choudhury, Kashif Shafique, Zaeema Ahmer.

**Methodology:** Tahir Turk, Anthony Wenndt, Moniruzzaman Bipul.

**Project administration:** Tahir Turk, Rina Paul, Syed Mahbubul Alam, Sohel Reza Choudhury, Kashif Shafique, Zaeema Ahmer, Anthony Wenndt, Hamidullah Khan Babar, Tannaza Sadaf, Moniruzzaman Bipul.

**Supervision:** Tahir Turk, Nilofer Fatimi Safdar, Nadia Shah, Syed Mahbubul Alam, Sohel Reza Choudhury, Kashif Shafique, Zaeema Ahmer, Anthony Wenndt, Hamidullah Khan Babar, Tannaza Sadaf, Moniruzzaman Bipul.

**Validation:** Rina Paul, Nilofer Fatimi Safdar, Nadia Shah, Anthony Wenndt.

**Writing – original draft:** Tahir Turk.

**Writing – review & editing:** Rina Paul, Nilofer Fatimi Safdar, Nadia Shah, Syed Mahbubul Alam, Sohel Reza Choudhury, Kashif Shafique, Zaeema Ahmer, Anthony Wenndt, Hamidullah Khan Babar, Tannaza Sadaf, Moniruzzaman Bipul.

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
