## [Decision Letter · Decision Letter 0]

19 Feb 2025

Dear Dr. Turk,

Thank you for submitting your manuscript to PLOS ONE. After careful consideration, we feel that it has merit but does not fully meet PLOS ONE’s publication criteria as it currently stands. Therefore, we invite you to submit a revised version of the manuscript that addresses the points raised during the review process.

We look forward to receiving your revised manuscript.

Kind regards,

António Raposo

Academic Editor

PLOS ONE

“This study was produced through the Global Alliance for Improved Nutrition (GAIN) Nourishing Food Pathways programme which is jointly funded by the German Federal Ministry for Economic Cooperation and Development; the Ministry of Foreign Affairs of the Netherlands; the European Union; the government of Canada through Global Affairs Canada; Irish Aid through the Development Cooperation and Africa Division (DCAD); and the Swiss Agency for Development and Cooperation (SDC) of the Federal Department of Foreign Affairs (FDFA).”

“Turk T, Shafik K, Choudhury S, Safdar N, and Mahbub-Alum S. are independent consultants to GAIN. Wenndt AJ, Babar HK, Sadaf T, and Bipul M work for GAIN as full-time employees. Other authors declare that they have no competing interest.”

4. Please remove your figures from within your manuscript file, leaving only the individual TIFF/EPS image files, uploaded separately. These will be automatically included in the reviewers’ PDF.

5. We note that this data set consists of interview transcripts. Can you please confirm that all participants gave consent for interview transcript to be published?

If they DID provide consent for these transcripts to be published, please also confirm that the transcripts do not contain any potentially identifying information (or let us know if the participants consented to having their personal details published and made publicly available). We consider the following details to be identifying information:

- Names, nicknames, and initials

- Age more specific than round numbers

- GPS coordinates, physical addresses, IP addresses, email addresses

- Information in small sample sizes (e.g. 40 students from X class in X year at X university)

- Specific dates (e.g. visit dates, interview dates)

- ID numbers

Or, if the participants DID NOT provide consent for these transcripts to be published:

- Provide a de-identified version of the data or excerpts of interview responses

- Provide information regarding how these transcripts can be accessed by researchers who meet the criteria for access to confidential data, including:

a) the grounds for restriction

b) the name of the ethics committee, Institutional Review Board, or third-party organization that is imposing sharing restrictions on the data

c) a non-author, institutional point of contact that is able to field data access queries, in the interest of maintaining long-term data accessibility.

d) Any relevant data set names, URLs, DOIs, etc. that an independent researcher would need in order to request your minimal data set.

For further information on sharing data that contains sensitive participant information, please see: https://journals.plos.org/plosone/s/data-availability#loc-human-research-participant-data-and-other-sensitive-data

If there are ethical, legal, or third-party restrictions upon your dataset, you must provide all of the following details (https://journals.plos.org/plosone/s/data-availability#loc-acceptable-data-access-restrictions):

1. A complete description of the dataset

2. The nature of the restrictions upon the data (ethical, legal, or owned by a third party) and the reasoning behind them

3. The full name of the body imposing the restrictions upon your dataset (ethics committee, institution, data access committee, etc)

4. If the data are owned by a third party, confirmation of whether the authors received any special privileges in accessing the data that other researchers would not have

5. Direct, non-author contact information (preferably email) for the body imposing the restrictions upon the data, to which data access requests can be sent

Reviewers' comments:

Reviewer's Responses to Questions

**Comments to the Author**

1. Is the manuscript technically sound, and do the data support the conclusions?

Reviewer #1: Yes

Reviewer #2: Yes

Reviewer #3: Yes

2. Has the statistical analysis been performed appropriately and rigorously?

Reviewer #1: Yes

Reviewer #2: Yes

Reviewer #3: Yes

3. Have the authors made all data underlying the findings in their manuscript fully available?

Reviewer #1: No

Reviewer #2: Yes

Reviewer #3: Yes

4. Is the manuscript presented in an intelligible fashion and written in standard English?

Reviewer #1: Yes

Reviewer #2: Yes

Reviewer #3: Yes

Reviewer #1: Your manuscript presents a comprehensive and innovative approach to integrating co-design into formative research for SBCC in nutrition-sensitive social protection programs. The methodology is robust, employing mixed methods and adhering to ethical standards. The use of the COREQ checklist and NVIVO software enhances the credibility of your qualitative analysis. Your focus on Bangladesh and Pakistan provides valuable insights for low- and middle-income countries. To strengthen your paper, consider elaborating on the practical implications of your findings and discussing any limitations. Additionally, providing more details on the results of the co-design sessions would further highlight the unique contribution of your research to the field.

Reviewer #2: The study presents a well-structured approach to addressing malnutrition through nutrition-sensitive social protection programs in Pakistan and Bangladesh.

While the use of semi-structured interviews and focus group discussions is commendable, the reliance on qualitative methods may limit the generalizability of findings.

The work involves a relatively small sample size (134 participants in focus groups and 30 interviews), which may not fully represent the diverse perspectives of all vulnerable populations. More extensive sampling across various demographics could enhance the validity and applicability of the conclusions drawn.

While co-design is beneficial for fostering ownership, it may also introduce bias if the participants are not representative of the broader population. The facilitation of these sessions should ensure that all voices, especially those of marginalized groups, are heard and valued.

The identification of barriers and opportunities is essential; however, the study could delve deeper into the underlying causes of these barriers. A more thorough exploration of systemic issues, such as socio-economic factors and policy constraints, could lead to more effective solutions.

While the paper emphasizes the importance of culturally nuanced messages, it would be valuable to provide more concrete examples of these messages and how they were developed. This would help in understanding the practical application of the research findings.

The manuscript mentions the development of an SBCC entry-point platform but does not address potential challenges in implementing the recommendations. It would be beneficial to discuss how these proposed interventions can be realistically integrated into existing social protection frameworks.

Finally, while the work highlights the immediate benefits of the research, there is a lack of discussion regarding the long-term evaluation and sustainability of the proposed SBCC interventions. Future research should consider how these programs can be monitored and adapted over time to ensure continued effectiveness in reducing malnutrition.

Reviewer #3: The Conceptual and Theoretical Framework

This study could be improved by providing a sound theoretical framework as the basis for the analysis. The study does not explicitly link the co-design with other established theories of behavioral change, such as health belief and TPB or social learning theory. The study could integrate a theoretical foundation to support SBCC interventions.

The assumption that co-design leads to better participation, but does not explicitly engage with limitations such as the issue of power dynamics in participatory research. Could you include examples of similar interventions on co-design that have been successful?

2. Methodological Weaknesses

The sample size of 30 SSI and 12 FDG could not have been enough to support the generalized findings of other LMICs

This should be acknowledged in the limitations.

The study does not explicitly explain how the participants were recruited or the sampling technique adopted.

It was also claimed that saturation was achieved, but no evidence was provided. Could it be coding framework stability or redundancy in responses?

What inter-coder reliability measure was employed?

Analysis and results

The thematic analysis is highly subjective, and it lacks quantitative validation. You could include basic frequency statistics to strengthen the themes.

The practical challenges, such as scalability and sustainability, were not addressed adequately despite the theoretical benefits of co-design.

The study used redundant phrases. For instance, “The needs assessments and integrated co-design sessions highlighted the benefits of close consultation with program beneficiaries in the design of culturally appropriate SBCC interventions to support nutrition-sensitive social protection programs”. This should be more concise.

Format the thematic analysis coding table properly

Discussion and Policy Recommendation

The study could be improved by being specific with the policy recommendation. SDGs were mentioned, but it does not explicitly detail how policymakers can effectively implement the findings.

Reference

The reference is poorly formatted.

**Do you want your identity to be public for this peer review?** For information about this choice, including consent withdrawal, please see our Privacy Policy

Reviewer #1: No

Reviewer #2: **Yes: ** M. João Lima

Reviewer #3: No

---

## [Author Response · Author response to Decision Letter 1]

7 Mar 2025

PLOS One Editor 27/08/2024

Dear Editor,

Thank-you for your review and feedback in regard to our manuscript: Integrating co-design into formative research for a SBCC entry-point platform for nutrition-sensitive social protection programs in low -and middle-income country settings. We have provided advice on how the issues have been addressed as follows:

Results and Discussion: Ensure that the full manuscript includes a comprehensive presentation of results and a thorough discussion of the findings.

Providing a comprehensive presentation of results for a complex study which includes three data sources – systematic review, KIIs and FGDs is a challenge. We have endeavored to present the results and discussion of key issues concisely, but have reviewed these sections and elaborated on the presentation of the results and discussion on the findings. Nevertheless, we have included additional sections on Results for Table 1. as follows:

“A fourth sub-theme to emerge was the lack of strategic approaches, which prevented the attainment of well designed, evidence-based SBCC programs with the collaboration of all stakeholders Other barriers mentioned by beneficiaries and stakeholders included poverty, widespread inequity, and gender issues including ‘gatekeepers’, predominantly the husbands of women who impeded their wives ability to successfully engage in social protection cash transfer and food programs as a result of distrust of any programs which may take their wives out of the household. Other sub-themes emerging that were seen to hinder program uptake were policy inertia, low literacy, substance abuse, and the distances and time required to travel to SPP recruitment offices/distribution hubs and access points (see Table 1. Barriers to Program Engagement).”

We also refined wording for Table 2, Table 3 and Table 4 to ensure each sub-theme was addressed and fully explained.

2. Data Availability: Include a statement about data availability in the full manuscript.

We have included the following advice in the manuscript after the conclusion in a header titled “DATA AVAILABILITY”

“The authors confirm that the data supporting the findings of this study are available within the article [and/or] its supplementary materials. Derived data supporting the findings of this study are also available from the corresponding author [TT] on request.”

Anonymized data-sets have been provided with the manuscript as supplementary files.

3. Implications: Elaborate on the practical implications of integrating co-design approaches in SBCC for nutrition-sensitive social protection programs.

The Conclusion chapter elaborates on the Implications section of the manuscript which precedes the Conclusion. It provides a number of key points including implications for future programs and lessons learned. We have reframed to be more specific.

4. Limitations: Discuss any limitations of the study and how they were addressed or might affect the interpretation of results.

The authors did include a limitations section in the manuscript as follows:

“A limitation of the study is that grounded-theory approaches to qualitative research can be heavily directed by a primary researcher, potentially leading to internal biases. Qualitative methods may also suffer from the lack of generalizability to broader populations given the smaller sample sizes used. Despite these limitations, the authors believe the study can still claim generalizability in relation to country findings given the adequate sample size and multiple data-sources from stakeholders and beneficiaries across both country settings. As such, the qualitative processes have also benefitted from substantiation through quantitative statistical measures analysed through the systematic review.”

However, we have now included a separate section in the manuscript titled “Limitations” and included the following text:

“Limitations of the study is that in moderation of FGDs, group dynamics and moderator influence—including unintentionally influencing participants' responses through moderator biases, body language, or leading questions—may result in skewed data that may not accurately reflect participants' true opinions. Other limitations include unequal participation or dominant participants in groups which may lead to the voices of most vulnerable participants being underrepresented in the discussions. Next, is the issue of maintaining group focus for study protocols which may take up to two hours to complete. Handling of sensitive topics around malnutrition and chronic diseases can also skew results, due to participants avoidance in discussing uncomfortable issues. Additionally, the complexity of the data, environmental influences, and the limited depth of discussions with all participants can also impact on the validity of the insights gathered. In some cases, the non-anonymity of focus group participants may inhibit individuals from sharing their true thoughts and feelings. Additional limitations of co-design approaches is that the process is likely more taxing to groups, given the range of stimuli being presented, and the complex number of issues covered. From a data analysis perspective grounded-theory approaches to qualitative research can also be heavily directed by a primary researcher, potentially leading to internal biases. Despite the limitations, the authors believe the study can still claim generalizability in relation to country findings given the adequate sample size and multiple data-sources from stakeholders and beneficiaries across both country settings. As such, the qualitative processes have also benefitted from substantiation through quantitative statistical measures analysed through the systematic review.”

5. Future Research: Suggest directions for future research based on the findings of this study.

We have included another header titled “Implications for future research” after the Limitations section. The following suggestions for future research are as follows:

“Given the potential for the integration of co-design into formative research stages of priority health programs, to provide more culturally specific solutions to SBCC program design, future studies should look at creative ways to include the beneficiaries of health programs into the design of the programs. This could include more opportunities for beneficiaries to contribute to the development of socially and culturally appropriate messages, creative approaches as well as providing advice on the most efficient dissemination channels for SBC messages and solutions given their lived experience of the problems that programmers are trying to address.”

We have also rewritten the Discussion Chapter to include more specific discussion on the themes emerging from the analysis and included a number of additional citations which concur with the findings of our study.

We hope we have been able to address the issues highlighted by PLOS One to the editor’s satisfaction.

Sincerely

Tahir Turk (MSc, PhD)

---

## [Decision Letter · Decision Letter 1]

26 Mar 2025

Integrating co-design into formative research for a SBCC entry-point platform for nutrition-sensitive social protection programs in low -and middle-income country settings

PONE-D-24-36509R1

Dear Dr. Turk,

We’re pleased to inform you that your manuscript has been judged scientifically suitable for publication and will be formally accepted for publication once it meets all outstanding technical requirements.

Kind regards,

António Raposo

Academic Editor

PLOS ONE

Additional Editor Comments (optional):

Reviewers' comments:

Reviewer's Responses to Questions

**Comments to the Author**

Reviewer #2: All comments have been addressed

2. Is the manuscript technically sound, and do the data support the conclusions?

Reviewer #2: Yes

3. Has the statistical analysis been performed appropriately and rigorously?

Reviewer #2: Yes

4. Have the authors made all data underlying the findings in their manuscript fully available?

Reviewer #2: Yes

5. Is the manuscript presented in an intelligible fashion and written in standard English?

Reviewer #2: Yes

Reviewer #2: Considering the major modifications made in the manuscript, I believe it can now be published since it's in an easy-to-read format.

**Do you want your identity to be public for this peer review?** For information about this choice, including consent withdrawal, please see our Privacy Policy

Reviewer #2: **Yes: ** M. João Lima

---

## [Editor Report · Acceptance letter]

PONE-D-24-36509R1

PLOS ONE

Dear Dr. Turk,

I'm pleased to inform you that your manuscript has been deemed suitable for publication in PLOS ONE. Congratulations! Your manuscript is now being handed over to our production team.

Kind regards,

on behalf of

Dr. António Raposo

Academic Editor

PLOS ONE